# Impact of Antibiotics Associated with the Development of Toxic Epidermal Necrolysis on Early and Late-Onset Infectious Complications

**DOI:** 10.3390/microorganisms9010202

**Published:** 2021-01-19

**Authors:** Bretislav Lipovy, Jakub Holoubek, Marketa Hanslianova, Michaela Cvanova, Leo Klein, Ivana Grossova, Robert Zajicek, Peter Bukovcan, Jan Koller, Matus Baran, Peter Lengyel, Lukas Eimer, Marie Jandova, Milan Kostal, Pavel Brychta, Petra Borilova Linhartova

**Affiliations:** 1Department of Burns and Plastic Surgery, Institution Shared with the University Hospital Brno, Faculty of Medicine, Masaryk University, Kamenice 753/5, 625 00 Brno, Czech Republic; bretalipovy@gmail.com (B.L.); holoubekjakub@yahoo.com (J.H.); dr.brychta@seznam.cz (P.B.); 2Central European Institute of Technology, Brno University of Technology, Purkyňova 656/123, 612 00 Brno, Czech Republic; 3Department of Clinical Microbiology, University Hospital Brno, Jihlavska 20, 625 00 Brno, Czech Republic; promarketa@email.cz; 4Institute of Biostatistics and Analyses, Faculty of Medicine, Masaryk University, Kamenice 753/5, 625 00 Brno, Czech Republic; cvanova@iba.muni.cz; 5Division of Plastic Surgery and Burns Treatment, Department of Surgery, Charles Univesrity, Faculty of Medicine and Teaching Hospital, Sokolská 581, 500 05 Hradec Králové, Czech Republic; leo.klein@fnhk.cz; 6Department of Military Surgery, Faculty of Military Health Sciences, University of Defence, Třebešská 1575, 500 01 Hradec Králové, Czech Republic; 7Prague Burn Center, Charles University, Faculty of Medicine and Teaching Hospital Kralovske Vinohrady, Srobarova 1150, 100 34 Prague, Czech Republic; iv.gross@yahoo.com (I.G.); zajicekrobert@seznam.cz (R.Z.); 8Department of Burns and Reconstructive Surgery, Faculty of Medicine, Comenius University in Bratislava, Ruzinovská 4, 821 01 Bratislava, Slovakia; bukovcan@ru.unb.sk (P.B.); koller@ru.unb.sk (J.K.); 9Department of Burns and Reconstructive Surgery, 1st Private Hospital Košice-Šaca, Lucna 512, 040 15 Košice, Slovakia; matus.baran@seznam.cz (M.B.); plengyel@nemocnicasaca.sk (P.L.); 10Department of Paediatrics, Charles University, Faculty of Medicine and Teaching Hospital, Sokolská 581, 500 05 Hradec Králové, Czech Republic; lukas.eimer@fnhk.cz; 11Department of Dermatovenerology, Charles University, Faculty of Medicine and Teaching Hospital, Sokolská 581, 500 05 Hradec Králové, Czech Republic; marie.jandova@fnhk.cz; 124th Internal Haematology Department, Charles University, Faculty of Medicine and Teaching Hospital, Sokolská 581, 500 05 Hradec Králové, Czech Republic; milan.kostal@fnhk.cz; 13Department of Pathophysiology, Faculty of Medicine, Masaryk University, Kamenice 753/5, 625 00 Brno, Czech Republic; 14Clinic of Stomatology, Faculty of Medicine, Masaryk University, Pekarska 664/53, 656 91 Brno, Czech Republic; 15Clinic of Maxillofacial Surgery, Institution Shared with the University Hospital Brno, Faculty of Medicine, Masaryk University, Jihlavska 20, 625 00 Brno, Czech Republic; 16Department of Molecular Pharmacy, Faculty of Pharmacy, Masaryk University, Palackeho tr. 1946/1, 61200 Brno, Czech Republic; 17Institute of Medical Genetics and Genomics, Faculty of Medicine, Masaryk University, Kamenice 753/5, 625 00 Brno, Czech Republic

**Keywords:** toxic epidermal necrolysis, antibiotics, infectious complication, early-onset infection, late-onset infection

## Abstract

Toxic epidermal necrolysis (TEN) is a rare disease, which predominantly manifests as damage to the skin and mucosa. Antibiotics count among the most common triggers of this hypersensitive reaction. Patients with TEN are highly susceptible to infectious complications due to the loss of protective barriers and immunosuppressant therapy. The aim of this study was to investigate the potential relationship between antibiotics used before the development of TEN and early and late-onset infectious complications in TEN patients. In this European multicentric retrospective study (Central European Lyell syndrome: therapeutic evaluation (CELESTE)), records showed that 18 patients with TEN used antibiotics (mostly aminopenicillins) before the disease development (group 1), while in 21 patients, TEN was triggered by another factor (group 2). The incidence of late-onset infectious complications (5 or more days after the transfer to the hospital) caused by Gram-positive bacteria (especially by *Enterococcus faecalis*/*faecium*) was significantly higher in group 1 than in group 2 (82.4% vs. 35.0%, *p* = 0.007/*p*_corr_ = 0.014) while no statistically significant difference was observed between groups of patients with infection caused by Gram-negative bacteria, yeasts, and filamentous fungi (*p* > 0.05). Patients with post-antibiotic development of TEN are critically predisposed to late-onset infectious complications caused by Gram-positive bacteria, which may result from the dissemination of these bacteria from the primary focus.

## 1. Introduction

Toxic epidermal necrolysis (TEN) and Stevens–Johnson syndrome (SJS) are representatives of so-called burn-like syndromes [1]. Although an infection, such as *Herpes simplex* virus, *Mycoplasma pneumoniae*, *Chlamydia pneumoniae*, etc., was identified as a trigger in some patients (parainfectious etiology) [2,3], the most common etiology of both these diseases is a drug-induced toxic-allergic reaction.

Currently, more than 200 preparations have been shown to have the potential to induce the development of these diseases; still, in approximately 5–10% of cases, no causative drug can be found in the patient history [4]. Antibiotics count among the most common triggers of this hypersensitive reaction [5].

The principal clinical manifestation of TEN is a skin exfoliation on at least 30% of the total body surface area (TBSA), caused by massive induction of apoptosis in the region of the dermo-epidermal junction [6,7,8]. Mucosal surfaces are often affected as well.

TEN pathophysiology has not yet been reliably explained. The current understanding assumes the reaction of the immune system and the interaction between the xenobiotic (e.g., drug or its metabolite) representing HLA antigens, which leads to subsequent clonal production of CD8+ T-lymphocytes (cytotoxic T lymphocytes) [9,10]. These drug-specific clones can then, through various mechanisms, initiate keratinocyte apoptosis. So far, no principal therapeutic concept has been universally accepted throughout all centers providing TEN treatment. Nevertheless, the application of immunosuppressants (corticosteroids, cyclosporin A, anti-TNFα) remains the most important systemic approach to the therapy of TEN patients [11,12,13,14,15,16,17,18].

TEN is a very rare disease, the annual incidence of which is, according to the available data, approx. 0.5–1 cases per million population [19,20,21]. The disease is burdened by high mortality, which is in some studies reported to be as high as 60%, although most current studies report much lower mortality of approx. 30% [19,22,23]. Considering the extensive loss of skin continuity, mucosal defects, and drug-induced immunosuppression during treatment, infectious complications represent (unsurprisingly) a dominant cause of mortality and morbidity in these patients.

Tracheobronchitis and pneumonia (or bronchopneumonia, respectively) count among the most common infectious complications in TEN patients. If this happens, especially if combined with an acute respiratory failure due to the mucosal involvement, it is necessary to perform bronchoscopy and laryngoscopy to verify the severity of the damage in the upper and lower respiratory tract [24]. Both TBAS (tracheobronchial aspirate) and bronchoscopy (with bronchoalveolar lavage—BAL, or protected specimen brush—PSB) can be used for diagnostic purposes. Other complications of TEN include urinary tract infection: patients may suffer from pyuria, and septic markers are usually elevated, which is accompanied by systemic signs of infection.

When diagnosing exfoliated wound infections (EWI), it is very important to supplement the qualitative evaluation with quantitative or, at least, semiquantitative evaluation using the surface imprint method. The basic cutoff value for distinguishing between wound colonization and wound infection is the concentration of ≥10^3^ colony-forming units (CFU)/gram tissue [25]. If the value lower, the bacterial presence is instead considered contamination or colonization, which is similar to the terminology used in patients with burns [26,27]. In TEN patients, it is, however, necessary to count with a rapid progression of the colonization and with the development of local infection, in particular, due to the compromised local defense mechanisms associated with the character of the disease itself or through drug-induced immunosuppression.

Diagnosis of infectious complications in TEN patients can often be a complex task. Clinical manifestations, as well as laboratory markers, can be partially affected by the character of the systemic and local therapy of these patients. In particular, administering immunosuppressants can influence the “traditional” symptomatology of the infection.

We assumed that the presence and/or character of infectious complications in patients suffering from an infection before the development of TEN (i.e., patients using antibiotics immediately before TEN development) would differ from those present in other TEN patients.

The aim of this project was to investigate a possible relationship between administering antibiotics (ATBs) immediately before TEN development and a risk of early or late infectious complications.

## 2. Materials and Methods

Patients with TEN diagnosis (L51.2 acc. to the MKN-10) hospitalized in burn centers or other departments in the Czech Republic and Slovakia (a catchment area of approximately 12.5 million population) in 2000–2015 were included in this study. The clinical and laboratory (histopathological) parameters listed below were used to establish the diagnosis.

The basic epidemiological, as well as specific, data (including the drug history prior to TEN development) were obtained from the international registry of the Central European Lyell syndrome: therapeutic evaluation (CELESTE) [25,28]. The study was performed with the approval of the Ethics Committee of the University Hospital Brno (11–-280617/EK, 28 June 2017).

### 2.1. Clinical Examination

A specific score of toxic epidermal necrosis (SCORTEN) scale [29,30] was used to determine the severity of the disease. The algorithm of drug causality for epidermal necrolysis (ALDEN) methodology [31] was used for the identification of drugs potentially acting as TEN triggers and patient stratification into groups (group 1: used antibiotics immediately before TEN development vs. group 2: no antibiotics were used immediately before TEN development).

The criteria for burn patients are usually used for diagnosis of infectious complications (sepsis, infection, bacteremia, bloodstream infection, tracheobronchitis, pneumonia, urinary tract infection, pyuria) in TEN patients as well [32]. The early-onset infection is defined as an infection developing within the first 5 days from the admission to the hospital, while infections developing after 5 or more days are considered late-onset infections. The same criteria were used for the patients in the CELESTE registry. When diagnosing pneumonia, the microbiological findings were correlated with X-ray positivity (new or progressive infiltrate, consolidation, or cavitation).

### 2.2. Microbial Analysis—Sampling and Cultivation

Imprints and swabs from exfoliated areas were repeatedly taken for microbial analysis, as were the tracheobronchial aspirate fluid (TBAS), sputum, or bronchoalveolar lavage (BAL), urine, and peripheral blood.

Primarily, the occurrence of bacteria, yeasts, and filamentous fungi in TEN patients was investigated. Several cultivation media were used: blood agar, chocolate agar, MacConkey agar (selective agar for Gram-negative microbes), blood agar-NaCl (selective medium for *Staphylococci* sp.), Wilkins Chalgren agar (for the culture of anaerobic microorganisms), URIselect (chromogenic medium for detection of both Gram-positive and Gram-negative microorganisms enabling detection according to the color change), and Sabouraud agar (selective medium for yeasts and filamentous fungi).

Culture media with the biological material were incubated for 18–24 h at 35–37 °C, blood agar and chocolate agar in the atmosphere with increased pCO_2_, anaerobic (Wilkins Chalgren) agar in the anaerobic atmosphere, remaining media in the normal atmosphere. If the cultivation was negative after 24 h, the cultivation was extended to 48–72 h. Blood culture tubes were cultured in an automatic system (Bactec, Becton Dickinson, Berkshire, UK) for 6 days. Sabouraud agar was incubated for 6 days at the temperature of 28–30 °C for yeast culture and for 7 days at room temperature for filamentous fungi culture.

The cultured microorganisms were identified using MALDI-TOF (Matrix-Assisted Laser Desorption/Ionization) based on the mass spectrum of the microorganisms.

When an invasive fungal infection was suspected, analysis of fungal antigens in the blood serum was performed, namely of the aspergillus antigen galactomannan (Platelia *Aspergillus* Ag, Bio-Rad, Hercules, CA, USA) and a fungal cell wall antigen 1,3-beta-D-glucan (Fungitell, Associates of Cape Cod Incorporated, East Falmouth, MA, USA) using the enzyme-linked immunosorbent assay (ELISA, ELx808, Dynex, Buštěhrad, Czech Republic).

#### 2.2.1. Imprints from the Exfoliated Surfaces

For semiquantitative assessment of the microbial colonization of the wounds, which allows observation of its dynamics in time, and thus the assessment of the effectiveness of both the local and systemic treatment, surface wound imprints were taken by placing strips of sterile filtration paper (5 × 5 cm) on the exfoliated surface and immediately transferred onto the blood agar. The medium was transported into the microbiological laboratory in the shortest time possible (0–2 h after sampling), where the sample was transferred to other cultivation media as well.

#### 2.2.2. Swabs from the Exfoliated Surfaces

Swabs from exfoliated surfaces (for anaerobic culture) were performed using sterile cotton swabs that were immediately after sampling immersed into the transport Amies medium (Amies, COPAN, Brescia, Italy). After transportation into the microbiological laboratory, the material was transferred onto the culture medium.

#### 2.2.3. Airway Samples

Samples from airways were collected during bronchoscopy. Biological material from the upper and lower respiratory tract was cultured on the aforementioned media; a microscopic mount was always prepared as well.

TBAS and sputum were homogenized using broncholysin (Mucobene, Ratiopharm, Ulm, Germany) in the ratio of 1:1, 5–10 sterile glass beads, and vortexing at 1050–1200 rounds per minute for 10 min. The homogenized aspirate fluid/sputum was further diluted using buffered saline up to 10^−3^, 10^−5^, and 10^−7^ dilutions. In the case of the BAL samples, 10 µL of were transferred on the culture media using a calibrated inoculating loop.

#### 2.2.4. Urine Samples

9 mL of urine were collected into a sterile test tube, and 1 μL samples were transferred on the culture media with a calibrated inoculation loop.

#### 2.2.5. Blood Samples

5–10 mL of peripheral blood were taken for culture, and immediately after sampling, they were inoculated into special blood culture tubes (BACTEC Plus Aerobic, BACTEC Lytic Anaerobic, Becton Dickinson, UK). The tubes were transported to the microbiological lab, and samples were analyzed using an automated culture device (Bactec, Becton Dickinson, UK). Where microorganism growth was detected (based on the increasing amount of CO_2_ associated with the growth), a microscopic mount was prepared, and one drop of the blood sample (approx. 100 µL of blood) was transferred onto the culture media.

### 2.3. Statistical Analysis

Due to the non-normal data distribution, the median and interquartile ranges were used to describe the continuous and ordinal variables. To be able to compare results with those published elsewhere (in which median values do not appear), we also included the mean and standard deviation (SD) in tables. Categorical variables are described using the counts and relative frequencies (%) of the patients in the defined groups.

Due to the violation of the normality assumption of the data, the nonparametric Mann–Whitney test was used to compare the probability distribution of continuous or ordinal variables between the two groups of patients. With respect to the low expected frequencies in the contingency tables, the independence of categorical variables was evaluated using Fisher’s exact test.

Bonferroni corrections for multiple analyses were applied according to the recommendation made by Perneger [33]. Corrections for multiple comparisons were made separately for individual infections, namely for four Gram-positive bacteria and seven Gram-negative bacteria. If the patients were repeatedly analyzed for the same hypothesis, i.e., in both early and late-onset complications, Bonferroni corrections were made as well. Each test comparing the occurrence of infections in group 1 and group 2 was corrected for two evaluations (early and late-onset infections).

The results of statistical tests are presented using exact *p*-values. The significance threshold was set to 0.05. For all statistical analyses, IBM SPSS Statistics ver. 23 and ver. 25 (IBM Corporation, 2015 and 2017, resp.) were used. The charts were created in Microsoft PowerPoint.

## 3. Results

### 3.1. General Epidemiological and Clinical Parameters

In all, data of 39 patients hospitalized in the study period with TEN (no SJS/overlap TEN patients were included in this study) were extracted from the registry (mean age ± standard deviation, SD, 51 ± 25 years). Eighteen patients used ATBs immediately before TEN development (group 1), while in 21 patients, no ATBs that could have caused the development of TEN were present in patients’ records (group 2). The most common triggers in group 2 were antiepileptic drugs (lamotrigine in 4 patients; phenytoin, valproate, carbamazepine, and phenobarbital in one patient each). Paracetamol was identified to be the inducing agent in two patients. Other medications associated, according to the ALDEN criteria, with TEN development were tetrazepam, allopurinol, sertraline, metamizole, and bendamustine. In 8 patients, no drug that could have induced TEN development was identified.

In group 1, the ATBs leading to TEN development were prescribed to treat the following infectious complications: Infections of the upper respiratory tract (5 patients), of the lower respiratory tract (4 patients), of the urinary tract (4 patients), and fever of unknown origin (5 patients).

The most common ATBs identified as possible triggers in our study were beta-lactams in 8 patients in total (5× aminopenicillins, 2× phenoxymethylpenicillins, 1× cefuroxime), followed by trimethoprim/sulfamethoxazole (TMP-SMZ) in 4 patients (3× alone, 1× in combination with vancomycin), and clindamycin (lincosamide) in 3 patients (2× alone, 1× in combination with ciprofloxacin). Other ATBs (norfloxacin, doxycycline, nitrofurantoin) were present in the history of one patient each. In 5 patients (27.8% of group 1), TEN most likely developed after the use of narrow-spectrum ATBs against Gram-positive cocci and in 1 patient (5.6% of group 1) after narrow-spectrum ATBs against Gram-negative rods. In the remaining 12 patients (66.7% of group 1), broad-spectrum ATBs were the possible triggers.

As a part of the treatment, corticosteroids (CS) were administered in 23 patients (60.5% of 38 patients) alone, CS in combination with intravenous immunoglobulins (IVIG) in 9 patients (23.7%), and a combination of CS with IVIG and cyclosporine A (CyA) was used for treatment in 6 patients (15.8%). No statistically significant difference between the character of the systemic immunosuppressant TEN therapy between groups 1 and 2 was detected.

No statistically significant differences between the two groups were found in the age of the patients, in the sex representation, in the extent of the exfoliated area, in SCORTEN, or in need for mechanical ventilation. The occurrence of mucosal involvement in group 1 (55.6%) was lower than in group 2 (85.7%); the difference was, however, borderline insignificant (*p* = 0.072).

A significant difference between groups was detected in the comparison of the median time from the development of the first symptoms and a transfer into the center providing appropriate specialized care (group 1:5 days vs. group 2:7 days *p* = 0.040), but not in the median length of the hospital stay (group 1:23 days vs. group 2:16 days, *p* > 0.05), or mortality (group 1:38.9% vs. group 2:23.8%, *p* > 0.05). Epidemiological and clinical data are presented in Table 1.

In the entire study group, regardless of the use of ATBs immediately before TEN development, we also analyzed the relationships between the above-mentioned parameters (age, sex, the extent of exfoliation, SCORTEN, mucosal involvement, and mortality) and the time of the transfer to a specialized center. A higher median SCORTEN value and mortality were detected in the group of patients with TEN who were transferred within 5 days from the onset of the clinical symptoms compared to those transferred later (SCORTEN 4 vs. 2, *p* = 0.056; mortality 47.1% vs. 18.2%, *p* = 0.082).

### 3.2. Infectious Complications

Infectious complications were present in 33 patients in total (84.6%). Although the risk of developing infectious complications was higher in group 1 than in group 2, the difference was not statistically significant (*p* > 0.05)—see Table 2. Nevertheless, infectious complications in patients with TEN caused by Gram-positive bacteria *Enterococcus faecalis*/*faecium* were associated with antibiotics used immediately before TEN development (*p* = 0.010/*p*_corr_ = 0.042).

No statistically significant relationship between the type of antibiotic/s (narrow-spectrum antibiotics against Gram-positive/negative bacteria, or broad-spectrum antibiotics) used immediately before TEN development and the infectious complications caused by individual pathogens was observed in the patients from group 1—see Figure 1 (*p* > 0.05). In both groups of patients with TEN, combined infectious complications caused by Gram-positive and Gram-negative bacteria, or even by those bacteria combined with yeasts/filamentous fungi were the most common (83.3% in group 1, 66.7% in group 2)—see Figure 1.

Most TEN patients suffered from both early-onset (<5 days after hospitalization) and late-onset (5 or more days after hospitalization) infectious complications (*n* = 25, 67.6%); in six patients, only early-onset infectious complications were recorded (16.2%). An isolated late-onset infection (i.e., without an early onset one) was found in two patients (5.4%). The incidence of late-onset infectious complications caused by Gram-positive bacteria was significantly higher in group 1 than in group 2 (82.4% vs. 35.0%, *p* = 0.007/*p*_corr_ = 0.014), while no statistically significant difference between groups was observed regarding infections caused by Gram-negative bacteria, yeasts, and filamentous fungi (*p* > 0.05)—see Figure 2.

## 4. Discussion

Antibiotics count among drugs most frequently associated with the development of SJS/TEN. The highest risk of inducing this toxic-allergenic reaction is observed for aminopenicillins, macrolides, quinolones, cephalosporins, etc. [34,35,36]. A special position in this respect is attributed to trimethoprim/sulfamethoxazole (TMP-SZM), a preparation with a relatively high-risk of potentiation of SJS/TEN development [37,38,39]. This fact was confirmed by a wide range of pharmacoepidemiologic studies. Antitubercular drugs could, due to the globally increasing prevalence of tuberculosis, become another group of drugs more frequently associated with SJS/TEN development in the near future [40,41].

The mechanism of the development of the disease itself has not been fully clarified yet. In 2008, Pichler presented a so-called PI (pharmaco-immunological) concept that is nowadays generally accepted [42]. It is based on the assumption that some drugs can, under certain circumstances, interact with the immune system (T lymphocytes) through several mechanisms—through direct interaction, as a prohapten or hapten. Direct interaction has been described only in a few antibiotic drugs; typical representatives are, e.g., penicillin G, forming covalent bonds on the lysine of endogenous proteins, or cephalosporins. Most drugs act as prohaptens, becoming haptens only after metabolization, in which cytochromes P450 play a key role (CYP450) [43,44]. The PI-concept actually defines the pharmacological interaction with immune receptors. This concept has been recently expanded by a genetic perspective [45].

One of the principal approaches in proactive therapy, as well as the approach of the “wait and see protocol“, which is currently only marginally used, is the immediate termination of the administration of any potentially hazardous substances that may have led to the initiation of the condition and have a potential to cause progression of the patient’s condition. This is, of course, not so easy under the circumstances of the therapeutic use of such a drug where its discontinuation comes with a risk of progression of the original disease. To this date, there is no recommendation clearly defining the approach to the substitution of the antibiotic drug, potentially triggering the SJS/TEN development. If the SJS/TEN occurs following an application of an antibiotic, further therapy must be controlled strictly by the results of the microbiological examination—prophylactic or empirical prescription is not recommended [46,47].

In our group of 39 TEN patients, the development of the syndrome was preceded by ATB administration in 18 patients (46.2%). Aminopenicillins (ampicillin or amoxicillin) were the most common drugs preceding TEN development (5 cases, 12.8%), followed by TMP-SMZ (4 cases, 10.3%, 3 alone, one in combination with vancomycin). All such ATBs identified in our registry are predominantly effective against the Gram-positive spectrum of bacteria. It could, therefore, be assumed that the ATBs should impact in particular early-onset infections in various compartments. The varying range of the spectrum of antimicrobial effects of these drugs poses a problem for our retrospective study; nevertheless, almost all these ATBs were effective (in particular) against the Gram-positive cocci, some of them also against Gram-negative bacilli.

One of the first and most important epidemiological studies systematically analyzing the influence of individual drugs on SJS/TEN was published in 1995 [48]. Roujeau et al. evaluated in their Severe cutaneous adverse reaction study (SCAR) a large database of approx. 120 mil. Inhabitants of several European countries. They identified, in total, 245 patients with SJS (89 patients), overlap TEN (76 patients) and TEN (80 patients) [48]. Results of their study are in agreement with ours as antibiotics were the most commonly identified possible triggers of TEN development (in 115 patients, 46.9%).

Sharma et al. published data analyzing high-risk drugs in SJS/TEN development in a group of 30 patients [49]. In total, 57 drugs were evaluated as risk-bearing, dominated by anticonvulsants (35.1%) and antibiotics (33.3%). Mockenhaupt et al. compared in their study, comprising data from international registries, risk ratios of individual drugs between two patient populations [36]. The first cohort included patients with the diagnosis of SJS, overlap TEN, and/or TEN (379 patients); the other cohort was free of those diagnoses (1505 patients). Antibiotics were drugs most often associated with the development of SJS/TEN or overlap TEN (29.6%). Levi et al. studied the drugs bearing a high-risk of SJS/TEN development in children [50]. Their study analyzed age-selected patients (≤15 years of age) from two previous multicenter studies (SCAR and the European case–control surveillance of severe cutaneous adverse reactions, EuroSCAR). They revealed a strong association between SJS/TEN development and the use of antibiotics in children.

In 2018, Sullivan et al. published the results of their poll, analyzing data from 251 former SJS/TEN patients (response rate of 5.6% out of 4500 addressed persons) [51]. The analysis of the available information revealed that almost a third of the patients used antibiotics prior to the SJS/TEN development (*n* = 81, 32.3%). Of these, TMP-SMZ was the possible dominant trigger (42 patients, 51.9%).

Papay et al. compared the reporting frequency of SJS/TEN resulting from the use of drugs in the US Food and Drug Administration adverse event reporting system (US FDA AERS) database and compared the results to the EuroSCAR study results as a reference to identify high-risk drugs [52]. The comparison of both databases identified, in all, 12 “highly suspect” drugs, including two representatives of antimicrobials, namely Nevirapine and TMP-SMZ. Antimicrobials have been reported as a significant trigger of various severe bullous diseases in many other publications as well [53,54,55,56,57].

Analysis of patients’ records in our study implied that patients in group 1 (i.e., using ATBs immediately before the development of SJS/TEN) were transferred into burn centers sooner than patients from group 2 (median 5 days vs. 7 days; *p* = 0.040). Many authors reported that a timely transfer into specialized centers has a significant positive influence on the mortality of TEN patients [58,59,60]. This was not confirmed in our patient population as the mortality was 47.1% in the patients who were transferred to the specialized center within 5 days, compared to 18.2% in the group of patients who were transferred to the specialized center after 6 days or more (*p* = 0.082). This was most likely caused by the fact that patients with the most severe course of the disease were brought to the specialized centers sooner than those in whom the course was less fulminant. The median score for toxic epidermal necrolysis (SCORTEN) of the patients who were transferred sooner was 4; in those who were transferred later, the median SCROTEN was 2 (*p* = 0.056). As mentioned above, high values of SCORTEN at admission correlate with the increased risk of death [28].

Infectious complications were recorded, in total, in 84.6% of patients with TEN; they occurred in almost all patients of group 1, while in group 2 (not using ATBs prior to the TEN development), only three out of four patients developed infectious complications (*p* > 0.05). This was most likely caused by the fact that the patients in group 1 already suffered from an infection before TEN development, and the treatment with the original ATB was insufficient due to the forced interruption. In the study by Tocco-Tussardi et al., at least one microbial infection was detected in 91.7% of patients with SJS/TEN [46].

The only statistically significant difference between groups 1 and 2 was detected in the case of late-onset infections caused by Gram-positive bacteria (*p* = 0.007/*p*_corr_ = 0.014). We could only speculate on the reasons for this observation—although this may be due to the original infection (as most community pathogens are Gram-positive) that has not been sufficiently treated due to the discontinuation of the ATB therapy after TEN development, we have no clear evidence supporting this speculation.

In view of the severity of the disease, it is not at all surprising that mixed microflora (Gram-positive and Gram-negative bacteria) was found in a vast majority of patients in our group. Infectious complications caused by yeasts and filamentous fungi were also more common in group 1. This was likely caused by the administration of broad-spectrum antibiotics representing an important risk factor for mycotic infections [61].

In our TEN patients, the most frequently identified potential pathogens from the Gram-positive spectrum were coagulase-negative *Staphylococci* (*Staphylococcus epidermidis,* 24 patients, 61.5%), *Enterococcus faecalis/Enterococcus faecium* (19 patients, 48.7%), and *Staphylococcus aureus* (11 patients, 28.2%). Of Gram-negative bacteria, *Pseudomonas aeruginosa* (17 patients, 43.6%), *Klebsiella pneumoniae* (12 patients, 30.8%), and *Escherichia coli* (11 patients, 28.2%) were the most common. A significant difference in the frequency of occurrence of infections by Gram-positive cocci, especially by *Enterococcus faecalis/faecium*, was observed between group 1 and group 2. Although *Enterococci* sp. constitute under normal circumstances a part of the normal gastrointestinal and urogenital microflora, they are currently becoming feared nosocomial pathogens due to the dramatic increase of their resistance to various antibiotics (vancomycin, linezolid, ciprofloxacin, and others), which is particularly true in the environment of critically ill patients [62,63].

Mahar et al. evaluated in their study infectious complications in 27 TEN patients [64] with the occurrence of such complications of 63%. *Staphylococcus aureus* was the most commonly identified pathogen (11 patients, 40.7%), followed by *Pseudomonas aeruginosa* (8 patients, 29.6%). A similar spectrum of pathogens (*Staphylococcus aureus*, *Pseudomonas aeruginosa*, *Enterobacteriaceae*) causing infectious complications in SJS/TEN patients was reported in other studies as well [65,66,67].

The results of our study imply that if the SJS/TEN develops in patients using ATB therapy, a progression of infectious complications can be expected. This is especially true if the ATB (as a possible trigger) is discontinued, which leads to the progression of the original infection. Moreover, the administration of immunosuppressant therapy can further support the development of infectious complications and their early dissemination from the original focus into additional locations, such as the exfoliated areas, lower respiratory tract, vascular bed, etc. Unfortunately, we are unable to compare this result with data from other studies as to the best of our knowledge; no study with a similar concept has been published so far.

Our study comes naturally with some limitations resulting from its retrospective and multicentric design. Furthermore, due to the rarity of this disease, the group of patients is relatively small. We perceive as another limitation that although we were able to acquire important data on the type of infectious disease in patients from group 1 before TEN development, the identification of specific pathogens was not possible. The principal reason is that the ATB therapy was, in a vast majority of cases, routinely prescribed by the general practitioner without any pathogen identification. We were, therefore, unable to determine with full certainty in our retrospective analysis whether or not the infectious complications after the TEN development were caused by the primary endogenous, secondary endogenous, or exogenous microflora. As this study provides the first analysis of TEN patients of this type, and with this concept, we are unable to compare the results with other authors.

## 5. Conclusions

Infectious complications have a major influence on the morbidity and mortality of TEN patients. In view of the nature of the disease and associated extensive exfoliation, mucosal damage, and drug-induced immunosuppression, these patients are highly susceptible to infectious complications. Patients who develop TEN after the use of ATBs are critically predisposed to late-onset infectious complications caused by Gram-positive bacteria, which may result from dissemination of the bacteria from the primary focus.

There are potential implications of our results into the clinical practice:As infectious complications increase the morbidity and mortality of TEN patients as well as the duration of the hospital stay and associated costs [68], it is strictly necessary to proactively search for potential sources of infectious complications (early detection of the pathogens);Where an infection was present before TEN development (typically where antimicrobials are a possible trigger), the pathogen must be reported when transferring the patient to a specialized department and addressed during treatment;At present, no generally accepted strategy on substitution of one antibiotic therapy with another in TEN patients is in place;Emphasis must be laid on rapid methods of identification and quantification of microorganisms with pathogenic potential (quantitative polymerase chain reaction if available); a combination with standard cultivation methods is, however, beneficial.Antibiotic drugs should be administered in a targeted way and only after sensitivity analysis (minimum inhibitory concentration—MIC, E-test, etc.);Antibiotics used as the substitution of the originally administered ATB drug should possess maximal differences both in the structure and metabolic pathway of action while, at the same time, being used in an approved indication;Biochemical analysis and individual parameters (C-reactive protein, procalcitonin, soluble CD14 subtype—sCD14-ST) play at present only a supplementary role in the TEN therapy, serving for monitoring of the effectiveness of antimicrobial treatment. Hence, far, there are no data on the possible use of these methods in SJS/TEN patients within the immunosuppression therapy;Prophylactic administration of antibiotics is strictly prohibited in patients with TEN developing after the administration of ATBs;Large multicentric studies are necessary to ensure sufficiently robust data.

## Figures and Tables

**Figure 1 microorganisms-09-00202-f001:**
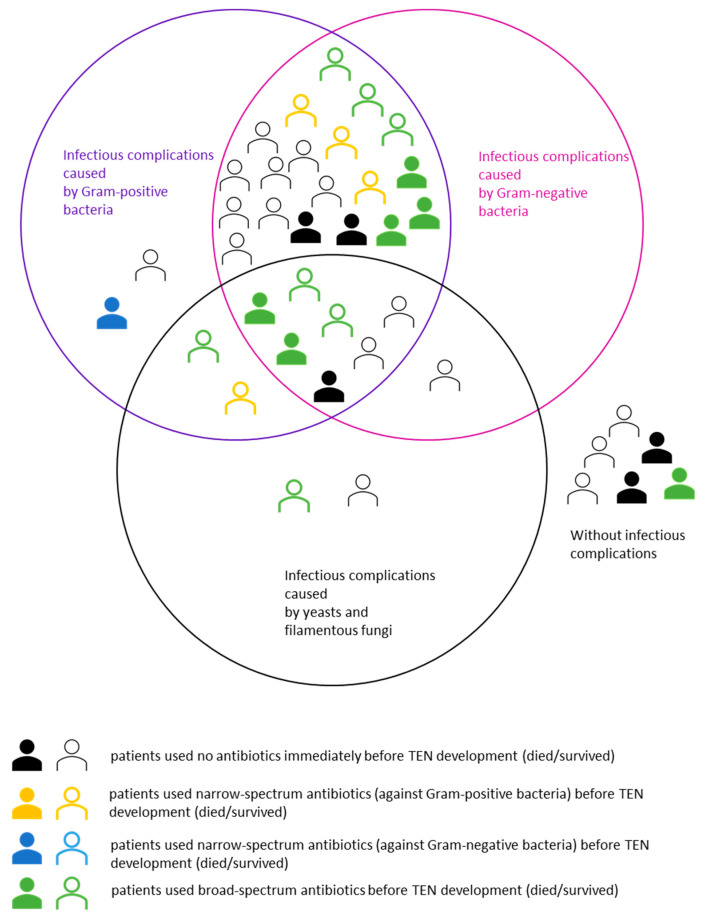
Clustering of 39 patients in the Central European Lyell syndrome: therapeutic evaluation (CELESTE) cohort according to the antibiotics used immediately before TEN development, subsequent infectious complications and their combinations, and patient survival.

**Figure 2 microorganisms-09-00202-f002:**
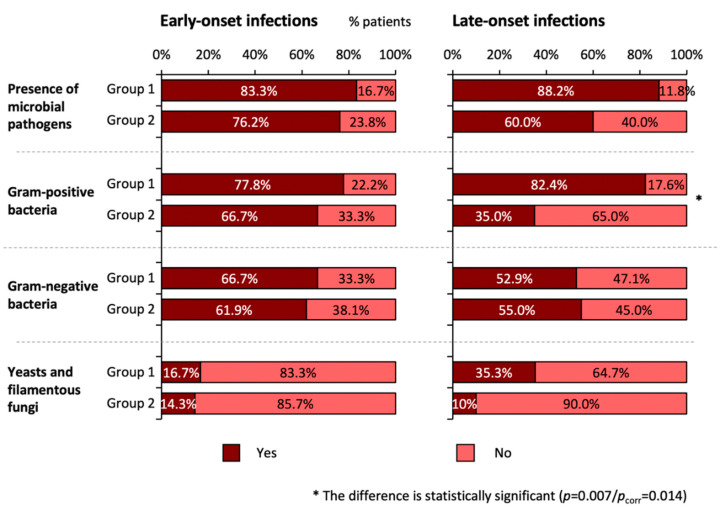
The incidence of early-onset (*n* = 39) and late-onset infectious complications (*n* = 37; 2 patients died less than 5 days after admission) in patients with toxic epidermal necrolysis (TEN) in the context of the use of antibiotics before TEN development. Fisher’s exact test was used for statistical evaluation. *P*_corr_ represents values after the Bonferroni correction for multiple analyses. Corrections were made for repeated analyses in the evaluation of early and late-onset infections. group 1: patients used antibiotics immediately before TEN development; group 2: patients used no antibiotics immediately before TEN development; early-onset, infectious complications developing within less than 5 days of admission to the hospital; late-onset, infectious complications developing 5 or more days after patient’s hospitalization.

**Table 1 microorganisms-09-00202-t001:** General epidemiological and clinical parameters of patients with toxic epidermal necrolysis (TEN) in the context of the use of antibiotics before TEN development.

Parameters	All Patients *n* = 39	Group 1 ^1^ *n* = 18	Group 2 ^1^ *n* = 21	*p*-Value ^2^
age (years), median (IQR)/mean ± SD	63 (30–69)/51 ± 25	65 (30–75)/54 ± 27	58 (30–66)/49 ± 23	0.426
sex (male), *n* (%)	16 (41.0)	7 (38.9)	9 (42.9)	1.000
extent of exfoliation (% of TBSA), median (IQR)/mean ± SD	70 (40–86)/67 ± 23	73 (50–89)/70 ± 23	70 (40–85)/65 ± 24	0.443
SCORTEN, median (IQR)/mean ± SD	3 (2–4)/3 ± 1	3 (2–4)/3 ± 1	3 (2–4)/3 ± 1	0.642
mechanical ventilation, *n* (%)	17 (43.6)	6 (33.3)	11 (52.4)	0.334
mucosal involvement, *n* (%)	28 (71.8)	10 (55.6)	18 (85.7)	0.072
time to transfer (days) ^3^, median (IQR)/mean ± SD	6 (5–8)/7 ± 4	5 (4–7)/5 ± 3	7 (5–12)/8 ± 4	0.040
LOS (days), median (IQR)/mean ± SD	18 (10–31)/21 ± 12	23 (12–34)/23 ± 13	16 (10–26)/19 ± 11	0.349
mortality, *n* (%)	12 (30.8)	7 (38.9)	5 (23.8)	0.488

SCORTEN, score for toxic epidermal necrolysis; TBSA, total body surface area; LOS, length of stay in hospital. ^1^ group 1: patients used antibiotics (ATBs) immediately before TEN development; group 2: patients used no ATBs immediately before TEN development. ^2^ For statistical testing of differences between group 1 and group 2, Mann–Whitney test was used for continuous parameters; for categorical parameters, Fisher’s exact test was used. ^3^ Days between the onset of the first symptoms of TEN and transfer to the specialized centers.

**Table 2 microorganisms-09-00202-t002:** A general overview of infectious complications in patients with toxic epidermal necrolysis (TEN) in the context of the use of antibiotics (ATB) before TEN development.

Parameters	All Patients *n* = 39	Group 1 ^1^ *n* = 18	Group 2 ^1^ *n* = 21	*p*-Value/*p*-Value_corr_^2^
Infectious complications, *n* (%)	33 (84.6)	17 (94.4)	16 (76.2)	0.190
Any Gram-positive bacteria, *n* (%)	30 (76.9)	16 (88.9)	14 (66.7)	0.139
*Staphylococcus aureus*	11 (28.2)	6 (33.3)	5 (23.8)	0.723/ns
*Streptococcus* sp.	8 (20.5)	2 (11.1)	6 (28.6)	0.247/ns
*Enterococcus faecalis/faecium*	19 (48.7)	13 (72.2)	6 (28.6)	0.010/0.042
Coagulase neg. *Staphylococcus*	24 (61.5)	14 (77.8)	10 (47.6)	0.098/ns
Any Gram-negative bacteria, *n* (%)	27 (69.2)	13 (72.2)	14 (66.7)	0.742
*Klebsiella pneumoniae*	12 (30.8)	5 (27.8)	7 (33.3)	0.742/ns
*Acinetobacter baumannii*	8 (20.5)	4 (22.2)	4 (19.0)	1.000/ns
*Stenotrophomonas maltophilia*	4 (10.3)	3 (16.7)	1 (4.8)	0.318/ns
*Pseudomonas aeruginosa*	17 (43.6)	7 (38.9)	10 (47.6)	0.748/ns
*Escherichia coli*	11 (28.2)	6 (33.3)	5 (23.8)	0.723/ns
*Enterobacter cloacae*	7 (17.9)	3 (16.7)	4 (19.0)	1.000/ns
*Proteus* sp.	8 (20.5)	4 (22.2)	4 (19.0)	1.000/ns
yeasts and filamentous fungi, *n* (%)	12 (30.8)	7 (38.9)	5 (23.8)	0.488

^1^ group 1: patients used ATBs immediately before TEN development; group 2: patients used no ATBs immediately before TEN development. ^2^ Fisher’s exact test was used for statistical evaluation. *p*-value_corr_ represents values after the Bonferroni correction for multiple analyses. Corrections were made for four Gram-positive bacteria and seven Gram-negative bacteria. ns, not significant.

## Data Availability

The data presented in this study are available on request from the corresponding author.

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
