# Peer review of "Impact of Antibiotics Associated with the Development of Toxic Epidermal Necrolysis on Early and Late-Onset Infectious Complications"

_microorganisms, 2021, doi:10.3390/microorganisms9010202_

Round 1

Reviewer 1 Report

N.A.

Author Response

Thank you.

Reviewer 2 Report

One may congratulate the authors on this study. It is novel and add new data to the current knowledge. To include 39 patients with TEN in one study is not easy. This is a rare disease as menntioned in the Introduction. I like the design of the reserach. The conclusions are important both from scientific and practical point of view.

I have only minor issue regarding mortality rate presented in the Introduction. I understand that searching the literature one may find the incidence up to 60%, but my experience is completely differnt one. Having the possibility of introducing the immunoglobulin therapy or immunnusupression (cyclosporine) the mortality rate is much lower. Therefore, I propose to change it or add comment to the text.

Author Response

Comment 1: One may congratulate the authors on this study. It is novel and add new data to the current knowledge. To include 39 patients with TEN in one study is not easy. This is a rare disease as menntioned in the Introduction. I like the design of the reserach. The conclusions are important both from scientific and practical point of view.

Answer 1: Thank you for your kind words.

Comment 2: I have only minor issue regarding mortality rate presented in the Introduction. I understand that searching the literature one may find the incidence up to 60%, but my experience is completely differnt one. Having the possibility of introducing the immunoglobulin therapy or immunnusupression (cyclosporine) the mortality rate is much lower. Therefore, I propose to change it or add comment to the text.

Answer 2: We thank the referee for this remark. The sentence was re-formulated in the text: "The disease is burdened by high mortality, which is in some studies reported to be as high as 60%, although most current studies report much lower mortality of approx. 30% [19,22,23]." 

This manuscript is a resubmission of an earlier submission. The following is a list of the peer review reports and author responses from that submission.

Round 1

Reviewer 1 Report

The manuscript by Lipovy and colleagues describes a cohort of TEN patients and compares infectious outcomes between those whose TEN was preceded by antibiotic use and those whose TEN was preceded by use of other drugs. The authors analysis of 39 patients suggested that those on antibiotics pre-TEN were more likely to develop gram positive infectious complications later.

A flaw in their approach is that the authors do not provide the spectrum of infections being treated in Group I with antibiotics prior to TEN onset. The reader cannot determine whether the organisms associated with later complications were causing disease beforehand. So the question remains whether the infectious complications were due to the same organism or due to loss of normal flora due to antibiotic use. The authors suggest the former explanation but without data to support their statement.

The authors also do not apply the statistical (Bonferroni) correction required by multiple analyses of the same data, and it is unclear whether the single statistically significant result they cite out of so many comparisons would remain significant.

Beyond those criticisms, the English grammar needs extensive editing, from the very first few sentences of the abstract onward.

Finally, the discussion' very lengthy description of drugs associated with onset of TEN is mostly irrelevant to the question posed by the article.

Reviewer 2 Report

General  comments :

It is a well-written and very interesting article. The study is well documented and refers to an interesting and previously unreported association between post-antibiotic development of TEN and  risk of early or late infectious complications. The authors provide detailed methodology and meaningful information about important parameters concerning the infectious complications in TEN patients, particularly in those with post-antibiotic development of TEN. Given that these serious complications may result from bacterial dissemination from primary infectious site, the dermatologists managed these patients should be aware of this possibility.   

Specific points to be addressed

  1. Results : Line 126 : Could you determine the factors of non-ATBs triggered TEN (in 21 patients) ?
  2.  2.  Clinical examination : line 353: Define the abbreviation ‘BSI’